# Key Indications for Passive Immune Prophylaxis Against SARS-CoV-2 Infection in Malignant Hematological Disorders: An Analytic Hierarchy Process by an Ad Hoc Italian Expert Panel

**DOI:** 10.3390/vaccines14010046

**Published:** 2025-12-30

**Authors:** Monia Marchetti, Giovanni Barosi, Francesco Passamonti, Marco Falcone, Emanuele Nicastri, Simona Sica, Pellegrino Musto, Francesca Romana Mauro, Corrado Girmenia

**Affiliations:** 1Hematology and Transplant Unit, Azienda Ospedaliera Universitaria di Alessandria, University of Eastern Pedemont, 15121 Alessandria, Italy; 2Center for the Study of Myelofibrosis, Scientific Direction, Istituto di Ricovero e Cura a Carattere Scientifico Policlinico San Matteo Foundation, 27100 Pavia, Italy; barosig@smatteo.pv.it; 3Hematology Division, Foundation IRCCS Ca’ Granda Ospedale Maggiore Policlinico, 20122 Milan, Italy; francesco.passamonti@unimi.it; 4Department of Oncology and Hemato-Oncology, University of Milan, 20122 Milan, Italy; 5Department of Clinical and Experimental Medicine, Azienda Osperdaliero Universitaria Pisana, University of Pisa, 56126 Pisa, Italy; marco.falcone@uniroma1.it; 6National Institute for Infectious Diseases Lazzaro Spallanzani IRCCS, 00149 Rome, Italy; emanuele.nicastri@inmi.it; 7Department of Laboratory and Hematological Sciences, Fondazione Policlinico Universitario A. Gemelli IRCCS, 00168 Rome, Italy; simona.sica@policlinicogemelli.it; 8Hematology Section, Department of Radiological and Hematological Sciences, Catholic University of the Sacred Heart, 20123 Rome, Italy; 9Hematology and Stem Cell Transplantation Unit, AOUC Policlinico, 70124 Bari, Italy; pellegrino.musto@blu.it; 10Department of Precision and Regenerative Medicine and Ionian Area, “Aldo Moro” University School of Medicine, 70121 Bari, Italy; 11Department of Translational and Precision Medicine, Sapienza University of Rome, 00185 Rome, Italy; mauro@bce.uniroma1.it; 12Dipartimento di Attività Cliniche Onco-Ematologiche, Dermatologiche e della Diagnostica Anatomo-Patologica, Azienda Ospedaliera Universitaria Policlinico Umberto I, 00161 Rome, Italy; girmenia@bce.uniroma1.it

**Keywords:** COVID-19, immune prophylaxis, vaccine, hematologic malignancies

## Abstract

Background: Pre-exposure passive immune prophylaxis (PrEP) might contribute to improve hematologic malignancy (HM) outcomes; however, there are currently no specific guidelines to inform patient selection. Methods: A literature review and a Delphi consensus process were used to identify COVID-19 risk factors, critical COVID-19 outcomes, and efficacy of PrEP against SARS-CoV-2 in HMs. An analytic hierarchy process was used to assign a priority score to candidate outcomes and to determine the PrEP indications. For these decisions, the experts assumed adequate compliance with anti-COVID-19 vaccination and acknowledged the effectiveness of PrEP in reducing SARS-CoV-2-related mortality and hospital admissions. Results: Based on the literature review, the expert panel identified 80 risk categories among patients with HM and prioritized eight clinical outcomes related to SARS-CoV-2 PrEP. The highest mean priority scores were observed for HM-related mortality (7.0), intensive care unit admission (6.7), and delays in anti-HM treatment (6.6). Based on such a framework, the experts deemed that if there was a variant-specific PrEP promptly available, it would be considered mandatory for all candidates receiving allogeneic hematopoietic cell transplantation, CAR-T therapy, or bispecific antibodies, regardless of local viral epidemiology. During epidemiological waves, variant-specific PrEP would also be recommended for patients with HMs at high risk of unfavorable COVID-19 clinical outcomes. Conclusions: This study identified PrEP indications for patients with HM receiving appropriate active immunization against COVID-19.

## 1. Introduction

COVID-19 still represents an active global public health concern, with ongoing transmission in most countries. Despite the rise in global vaccination rates and natural immunity, SARS-CoV-2 infection continues to pose clinical challenges for vulnerable populations [1]. According to the latest update from the World Health Organization (WHO), SARS-CoV-2 is most likely transitioning towards endemicity in some regions of the world, particularly in areas with low vaccination coverage or waning immunity [1].

An increase in the SARS-CoV-2 activity with the emergence of new Omicron-descendant variants and a rise in infections in some countries worldwide have been reported by the World Health Organization (WHO) and the European Centre for Disease Prevention and Control (ECDC) [2,3,4].

The SARS-CoV-2 pandemic and subsequent epidemic waves have caused severe respiratory infections in many immunocompromised individuals despite vaccination and antiviral therapies [5]. In particular, individuals with hematologic malignancy (HM) have experienced the highest rates of COVID-19-related hospitalization and mortality [6,7,8,9,10,11,12,13,14,15,16,17]. A major reason of the severe clinical impact of COVID-19 in HM individuals was their low response to COVID-19 vaccines, with a 67.7% overall seroconversion rate, lower than that in both the general population and in individuals with solid cancers [18]. Despite the declined virulence of the latest circulating SARS-CoV-2 variants, Omicron-related waves still determine negative clinical outcomes in patients with HM. In a Beijing study, 10.4% of 412 HM patients developed COVID-19 pneumonia, with severe cases largely reported among those with advanced malignancies [19].

In this evolving epidemiological scenario, pre-exposure passive immune prophylaxis (PrEP) with anti-spike monoclonal antibodies aims to provide HM patients with an effective prolonged prophylaxis against SARS-CoV-2 infections [5]. Although global data on the adoption of PrEP are currently unavailable, there is considerable variation in decision-making among individuals considering PrEP [17]. Key factors contributing to uncertainty regarding PrEP strategy in HM include the limited susceptibility of recent SARS-CoV-2 variants to PrEP according to different epidemiological scenarios [20,21]. Although 2023 and 2024 have seen declining deaths, hospitalizations, and intensive care unit (ICU) admissions compared to 2020–2022, COVID-19 continues to pose a significant global health risk in HM individuals.

Given this context, a project was implemented to develop guidelines for the use of SARS-CoV-2 PrEP strategies in patients with HM, with the goal of supporting improved COVID-19 and HM outcomes through specific indications.

## 2. Methods

The range of individuals with HM who could benefit from PrEP and the strength of PrEP indications were defined by a consensus process that started in June 2024. The panel of experts was composed of seven senior hematologists and two specialists in infectious disease selected based on their relevant knowledge, interest, and skill in COVID-19 disease.

During an initial meeting, the panel agreed on the project’s goal: to grade the strength of the indications of PrEP use in patients with HM stratified according to different SARS-CoV-2 risks, considering appropriate vaccine coverage and the effective capability of PrEP in reducing SARS-CoV-2-related hospitalization and mortality [14].

An extensive literature search was conducted to collect widely recognized risk factors for COVID-19 and assess the clinical impact of PrEP in these patients. Two independent members of the expert panel based on the electronic databases Scopus, PubMed, and Embase selected papers written in English from database inception to November 1st, 2024, adequately representing the available evidence. The first query was addressed to define the different risks of SARS-CoV-2 infection in patients with HM and was initially restricted to meta-analyses. The second query assessed the effects of PrEP in these patients. Common MeSH terms and free text words that were used to search for relevant publications in the field were as follows: Query n.1: ‘coronavirus disease 2019’ AND (‘COVID-19’:ti OR ‘SARS-CoV-2’:ti) AND (lymphoma OR leukemia OR myeloma OR thrombocyt* OR myeloprol*) AND ‘meta analysis’/de [166 records]. Query n.2: (‘cilgavimab plus tixagevimab’ OR ‘pre-exposure prophylaxis’) AND ‘coronavirus disease 2019’ AND (‘COVID-19’:ti OR ‘SARS-CoV-2’:ti) AND (lymphoma OR leukemia OR myeloma OR thrombocyt* OR myeloprol*) [206 records]. Additionally, abstracts presented at the 2024 Meeting of the American Society of Hematology were directly searched through the open-source website (https://ash.confex.com/ash/2024/webprogram/start.html accessed on 1 November 2024). Overall, 10 abstracts were selected based on the original data reported.

To define the strength of indication for PrEP, the analytic hierarchy process (AHP)—a multi-criteria decision-making method used to structure complex problems and make informed decisions—was used [22]. The AHP involved the expert panel guided by a methodologist (MM) who delivered and analyzed three online anonymous questionnaires using a modified Delphi technique. The members of the panel first ranked eight clinical outcomes of SARS-CoV-2 prophylaxis and selected the three most important ones. Subsequently, the panel listed 80 risk classes defined by host–disease–therapy combinations. In a subsequent round, the members of the panel ranked how much the critical SARS-CoV-2 prophylaxis outcomes were relevant in each of the risk classes. The relevancy of SARS-CoV-2 prophylaxis was judged using a 5-level Likert scale and an equivalent “relevance score” ranging from 1 to 5. The portion of panelists judging COVID-19 PrEP as “relevant” or “absolutely relevant” was calculated for each of the 80 risk classes. Additionally, a “standardized ratio” (sr) between the arithmetic mean and the standard deviation was estimated. Based on these estimations of clinical relevance, the expert panel agreed to classify patients into three major risk classes (Figure 1). The highest-ranked class included those conditions achieving an sr higher than 5 and scored clinically relevant by more than 75% and absolutely relevant by more than 40% of the panelists.

## 3. Results

### 3.1. Review of the Evidence

The first evidence review found that among host-related risk factors, age and co-morbidities had the most robust and well-supported studies (Appendix A). However, age categories and co-morbidity assessments varied across studies. The presence of three or more co-morbidities was frequently identified as a strong predictor for severe SARS-CoV-2 infection outcomes in several disease subgroups by the EPICOVIDEHA international survey and other studies [5,9,13,15]. Specific co-morbidities such as diabetes, cardiopathy, and renal failure were also often reported as independent predictors for ICU admission and mortality. Surrogate markers of immunodeficiency, such as lymphopenia, were only partially addressed by the available literature and therefore only partially supported as independent risk factors.

Among disease-related risk factors, lower seroconversion rates and worse COVID-19-related outcomes were described in patients with several lymphoid neoplasms (Appendix A). Indolent lymphoproliferative neoplasms, such as smoldering multiple myeloma (MM) or untreated indolent non-Hodgkin lymphoma (NHL), and Hodgkin disease (HD) were not associated with unfavorable outcomes after SARS-CoV-2 exposure. Heterogeneous COVID-19-related outcomes were reported in patients with myeloid disorders (Appendix A): in particular, chronic myeloid leukemia (CML) patients showed favorable outcomes during infection, while acute myeloid leukemia (AML) and myelofibrosis were associated with high COVID-19-related fatality rates.

Finally, therapy-related risk factors were extensively explored (Appendix A). Clinical remission was closely associated with a significant decline in COVID-19-related mortality. A substantial body of high-quality consistent evidence supported a close relationship between worse seroconversion and poor outcomes in candidates and recipients of cellular therapies and some specific pharmacologic therapies, such as anti-CD20 and anti-CD19 monoclonal or bispecific antibodies, JAK2 inhibitors, and BTK and BCL2 inhibitors. In particular, patients who underwent allogeneic hematopoietic cell transplantation and CAR-T recipients were more likely to develop a low or absent serological response compared to the healthy control group (OR inferior to 0.25).

The second systematic literature search we made retrieved a total of 40 studies (more than 9000 reported patients) consistently demonstrating the capability of PrEP to significantly reduce SARS-CoV-2-related mortality and hospitalization rates. Four meta-analyses addressed the efficacy of tixagevimab/cilgavimab (T/C) in immunocompromised individuals (Appendix A). The literature review, while showing a 13.1% breakthrough COVID-19 rate with 14.9% requiring hospitalization, demonstrated a low rate of ICU hospitalization (2.6%) and mortality (3.4%). However, the association between PrEP use and treatment delays for underlying hematologic diseases was not investigated by any published study (Appendix A).

### 3.2. Grading of Critical Outcomes

The comprehensive Delphi consensus technique provided a list of eight outcomes that were judged to be critical for PrEP indication. The panel agreed that delay in HM treatment, ICU admission, and death were considered unfavorable outcomes that could be prevented by PrEP strategies (Table 1). Treatment delays were considered especially relevant during periods of no active SARS-CoV-2 circulation.

The panelists subsequently ranked the clinical relevance of COVID-19 prophylaxis in different patient groups based on three main outcomes: (1) severe clinical impact of treatment delay caused by COVID-19, (2) higher rate of severe COVID19 and ICU admission during COVID-19, and (3) poor seroconversion after SARS-CoV-2 vaccination.

Therefore, the panelists ranked the clinical relevance of COVID-19 prophylaxis across 80 risk classes (Table 2 and Appendix A), and among them, they identified three major rank classes.

### 3.3. Indications for PrEP

Based on the assigned ranks (Table 2), the following indications for the use of PrEP in patients with HM were demonstrated (Figure 2):Due to potential treatment delay due to COVID-19, passive SARS-CoV-2 immunoprophylaxis was strongly recommended (mandatory) for individuals with HM who were candidates for allogeneic HCT within 3 months or for CAR-T therapy. Passive immunoprophylaxis was also considered mandatory for patients with advanced-phase HM eligible for therapy with bispecific antibodies. These recommendations were deemed applicable regardless of the epidemiological spread of the SARS-CoV-2 virus variants.Passive immunoprophylaxis was additionally recommended for individuals at elevated risk of developing severe COVID-19 during waves of SARS-CoV-2 driven by more virulent variants. This group includes patients with high-risk myeloid neoplasms, older patients with multiple myeloma undergoing active therapy, and patients with lymphoid neoplasms who have profound humoral or cellular immunosuppression, such as those affected by CLL.The panel agreed on recommending that in HM patients the risk of SARS-CoV-2 infection should be periodically re-evaluated to assess the priority level for the need of SARS-CoV-2 passive immunoprophylaxis.

## 4. Discussion

During the first two waves of the SARS-CoV-2 pandemic, the infection-related fatality rate in patients harboring HM was 41 times higher than that of the general population [15], and even in the subsequent epidemic phase with less pathogenic Omicron variants, the outcome for HM patients continued to be less favorable than that of the general population [18,19]. One of the reasons for the high mortality recorded in patients with HM is their low response to the active immune prophylaxis against the SARS-CoV-2 virus. A metanalysis, which included 150 studies that analyzed a total of 20,922 HM patients, revealed a 68% seroconversion prevalence rate following SARS-CoV-2 vaccination [23], 30% lower than that reported in solid cancer patients [23,24,25,26,27]. Although the seroconversion rate improved after further vaccine doses [28], it remained unsatisfactory even after boosters [23], particularly in the post-transplant phase for fully vaccinated individuals [29].

Based on evidence regarding the benefits of variant-specific PrEP passive immunization in HM patients, including those who had completed a full vaccination course, this consensus exercise was conducted to provide recommendations for a PrEP strategy in HM. The consensus panel defined recommendations aimed at providing guidance on COVID-19 prevention where evidence is currently limited or uncertain. Given the complex, multidisciplinary nature of the topic, we recognized the need for a thorough assessment of COVID-19 prevention strategies, making PrEP indications an ideal subject for a consensus study. The approved indications support the PrEP option strategies, considering the complex nature of the disease. Utilizing a transparent, motivation-based analytic hierarchy process (AHP), our multidisciplinary panel identified patient groups eligible for PrEP allocation, both during periods of low viral prevalence and throughout SARS-CoV-2 peak waves.

The criteria for using PrEP as a mandatory or recommended strategy in HM patients are based on the risk of severe SARS-CoV-2 infection and the timing required when therapeutic deferral is not feasible due to background clinical co-morbidities. The use of PrEP should be tailored to the current epidemiological context. Death, severe infection (such as ICU admission), and delays or downgrades in hematologic disease treatment due to COVID-19 are critical factors affecting PrEP’s risk–benefit assessment. Our extensive literature review demonstrates the effectiveness of PrEP, in addition to full vaccination, in reducing severe COVID-19 infections and death in various HM patient populations. The prevention of infections that may interfere with the therapeutic management of potentially life-threatening HMs had significant clinical consideration. Although explicit data are lacking, we believe that preventing severe SARS-CoV-2 infection before urgent hematologic treatment is highly relevant to patient outcomes.

Based on these criteria, we consider PrEP mandatory for allogeneic HCT, CAR-T, and bispecific therapy candidates, regardless of virus epidemiology scenarios. In such cases, COVID-19 presents a serious risk—not just due to the infection itself but also because it may delay essential treatments where timing is critical to patient outcomes. Conversely, in other patient populations at high risk for severe viral disease due to immunological impairment, co-morbidities, and probable poor response to active vaccine immunization, PrEP pharmacology strategies may be adjusted during periods of high epidemic diffusion.

The population-based prevalence of viral variants exhibiting decreased susceptibility to monoclonal antibody-mediated neutralization substantially influences the clinical rationale for PrEP use. PrEP with T/C was reported to be highly effective when susceptible SARS-CoV-2 variants, such as BA.1, were prevalent, with a low rate of breakthrough infections (11–16%) and a low case fatality rate (1–2%) [27,28]. Higher breakthrough rates occurred when non-susceptible variants like BA.5 and XBB were most common [30,31,32]. In view of the high prevalence of SARS-CoV-2 variants non-susceptible to the use of T/C, PrEP was withdrawn from clinical use by the US Food and Drug Administration in January 2023 [33].

The new monoclonal antibody sipavibart was evaluated in the SUPERNOVA randomized controlled trial, which tested the efficacy of this anti-spike monoclonal antibody for PrEP of COVID-19 in immunocompromised individuals [34,35]. The trial was successful, showing a relative risk reduction of 34.9% in symptomatic COVID-19 due to any SARS-CoV-2 variant compared to the control arm, who received T/C. However, the efficacy of sipavibart declined in the final months of the SUPERNOVA study due to the emergence of variants with a spike mutation at Phe456Leu, which led to full resistance to sipavibart in vitro. Of note, by January 2025, Phe456Leu variants (including KP.2, KP.3, and XEC) accounted for 96% of the sequences deposited in the GISAID database [36]. While the phenomenon of the worldwide diffusion of SARS-CoV-2 variants with spike antigenic change leading to resistance to T/C was observed several months after the clinical use of the monoclonal antibody, with sipavibart, the emergence of resistant Phe456Leu variants occurred while it was being tested, illustrating how quickly and unpredictably a passive immunization COVID-19 prevention strategy can lose clinical benefit.

Recent data show the neutralizing activity of another monoclonal antibody, pemivibart, against currently dominant SARS-CoV-2 variants [37]. The recently updated Clinical Practice Guidelines by the Infectious Diseases Society of America on the Treatment and Management of COVID-19 in immunocompromised individuals at risk of progression to severe COVID-19 suggest pre-exposure prophylaxis with pemivibart when predominant regional variants are susceptible to the agent [38].

Some limitations of the present study need to be reported. In particular, the negative clinical consequences of treatment delay due to COVID-19 in patients with hematologic malignancies is a soft outcome rarely reported by the published studies. Therefore, we claim the need for clinical studies targeting specific outcomes of COVID-19 prevention in oncologic patients. Moreover, the small size of the panel represented a partial burden for the AHP; therefore, the risk scores of some classes partially overlapped.

## 5. Conclusions

In conclusion, by using an AHP-based, literature-driven consensus, we propose criteria for optimizing the use of PreP in patients with HM, both in a low-SARS-CoV-2-prevalence timeframe and during significant epidemiological waves. These criteria should be valid as long as a good portion of the circulating viral strains maintain in vitro susceptibility to the prophylactic monoclonal antibody.

The available evidence suggests that, in addition to the vaccination strategy, a variant-specific and promptly available PrEP would be the optimal approach to contain the effects of COVID-19 in immunocompromised patients. However, further data on ongoing research into monoclonal antibodies (mAbs) with improved antiviral stability and efficacy is still needed [39]. The identification of new immunogenic epitopes against the SARS-CoV-2 spike protein are of crucial importance for the development of next-generation antibodies with resilience to viral evolution that are suitable for preventive or therapeutic approaches against COVID-19 [40,41,42]. Continuous virological surveillance is pivotal to detecting any emerging variants with reacquired sensitivity to available antiviral molecules.

## Figures and Tables

**Figure 1 vaccines-14-00046-f001:**
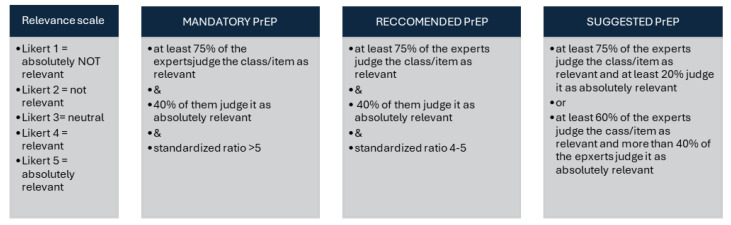
Analytic hierarchy process. Standardized ratio is the ratio between the mean and standard deviation of the relevance score. The standardized ratio enhanced the agreement among the experts.

**Figure 2 vaccines-14-00046-f002:**
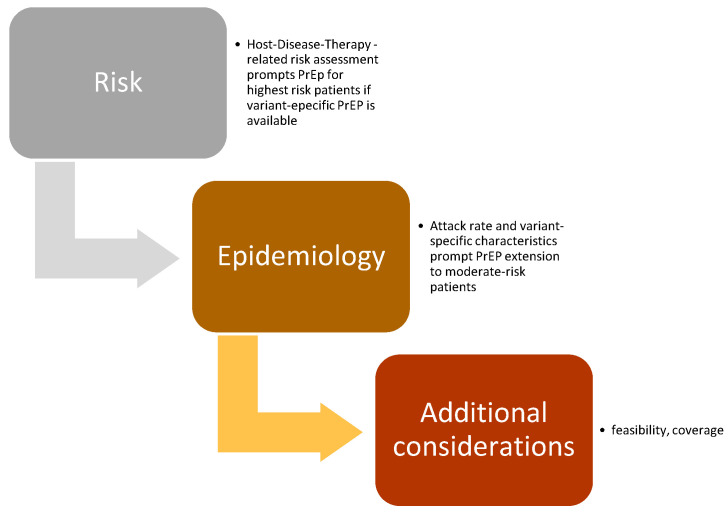
PrEP recommendations: interaction between SARS-CoV-2 epidemiology and patient risk profile.

**Table 1 vaccines-14-00046-t001:** Ranking the clinical relevancy of COVID-19 outcomes in patients with blood disorders.

Outcome	Aruthmetic Average	Geometric Average
Death	7.00	7.00
ICU admission	6.75	6.74
Hematologic treatment delay	6.63	6.61
Hospital admission	6.25	6.24
Treatment downgrading	6.13	6.07
Virus persistence	5.13	4.88
Infection	4.75	4.48
Severe Adverse Events related to PrEP	4.25	3.87

**Table 2 vaccines-14-00046-t002:** Relevance of COVID-19 prophylaxis according to risk classes.

Therapy- or Disease-Related Risk Factors	Standardized Relevance Score (sr)	Rank Class
Candidate for allogeneic HSCT	6.6–9.7	Class I
Candidate for CAR-T (ALL, NHL, MM)	3.7–6.6	Class I
Candidate for (or ongoing) bispecific antibodies (NHL, MM)	3.7–4.3	Class I
After CAR-T After allogeneic HSCT	2.9–3.7 3.5–4.6	Class II
Ongoing anti-CD20 or CD19 Moab	3.3	Class II
BTKi or BCL2i (CLL, NHL)	3.2–5.1	Class II
Chemoimmunotherapy (NHL)	3.2–3.5	Class II
AML (or high-risk MDS) induction therapy	2.1–3.3	Class II–III
JAK2i, TKI, IMIDs	1.7–3.2	Class II
Low-risk MDS on therapy	2.1–2.3	Class III
Multiple myeloma	2.0–2.2	Class III
CLL (on Igiv prophylaxis)	2.3 (3.5)	Class III (II)
Immune cytopenias (prior splenectomy)	2.1–2.8 (2.0)	Class III
Bone marrow failure syndromes (on complement inhibitor therapy)	2.3 (2.3)	Class III

Legend: MM = multiple myeloma; NHL = non Hodgkin’s lymphoma; ALL = acute lymphoblastic leukemia; CLL = chronic lymphocytic leukemia; AML = acute myeloid leukemia; TKI = tyrosin kinase inhibitors; JAK2i = JAK2 inhibitors; BTKi = BTK inhibitors; BCL2i = BCL2 inhibitors; Moab = monoclonal antibody; CAR-T = chimeric antigen receptor T cells; MDS = myelodysplastic syndrome; IMID = immunomodulating drug. Rank class I: sr > 4.0; Rank class II: sr 3.0–3.9; Rank class III: sr < 3.0.

## Data Availability

No new data were created or analyzed in this study. Data sharing is not applicable to this article.

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
