# Peer review of "Key Indications for Passive Immune Prophylaxis Against SARS-CoV-2 Infection in Malignant Hematological Disorders: An Analytic Hierarchy Process by an Ad Hoc Italian Expert Panel"

_vaccines, 2025, doi:10.3390/vaccines14010046_

Round 1
Reviewer 1 Report
Comments and Suggestions for Authors
The authors have presented a valuable and timely AHP-based consensus on PrEP indications for hematologic malignancies, yet several refinements can improve the paper:
-Authors need to clarify methodological transparency in the AHP steps, especially how the 80 risk classes were operationalized and how consistency ratios were ensured.
- They should provide stronger linkage between literature evidence and final recommendations, some jumps in logic feel fast, especially in the Discussion.
-They can address the limitation that evolving variant resistance may rapidly invalidate PrEP recommendations and highlight how the framework adapts dynamically.
-They need to expand on how treatment-delay data (largely absent in literature) were inferred, this assumption is central but underdeveloped.
Author Response
The authors have presented a valuable and timely AHP-based consensus on PrEP indications for hematologic malignancies, yet several refinements can improve the paper:
Comment 1. Authors need to clarify methodological transparency in the AHP steps, especially how the 80 risk classes were operationalized and how consistency ratios were ensured.
Response 1.
- We improved AHP transparency by specifically reporting the operationalization process into Supplementary Table 4 and add-on Supplementary Table 5
- IDSA guidelines (doi: 10.1093/cid/ciaf424.) recently suggested PrEp in immunocompromised individuals and highlighted high risk patients including namely recipients of stem cell transplants (<2 years), B-cell depleting agents in past 12 months (eg, rituximab, ofatumumab, ocrelizumab), CART-cell therapy (<12 months). Patients receiving Tyrosine kinase inhibitor (eg, ibrutinib, acalabrutinib) were assigned to moderate risk class
- We ameliorated the transparency of standardized ratios by adding a Legend to Figure 1 reporting “standardized ratio” definition and goal.
Comment 2. They should provide stronger linkage between literature evidence and final recommendations, some jumps in logic feel fast, especially in the Discussion.
Response 2. The evidence-to-recommendation process could not be strictly adherent to GRADE standards, because of several underreported outcomes, e.g. treatment delay, and to inconsistencies among COVID-19 waves. Therefore, the authors agreed with the reviewer that “evidence-based” was not an adequate label for the project: we delete this label from the manuscript Title.
Comment 3. They can address the limitation that evolving variant resistance may rapidly invalidate PrEP recommendations and highlight how the framework adapts dynamically.
Response 3.
- Effectiveness and timeliness of PrEP availability is not granted for future COVID19 variants. Therefore, we reminded the “hypothetical” role of our recommendations, “provided that an effective variante-specific PrEP is available”.
- Recently published evidence further supports the effectiveness of variant-adapted PrEP[1]. Indeed IDSA recently recommended Pemivibart pre-exposure prophylaxis to most of immunocompromised individuals “because they have the highest risk of inadequate immune response and progression to severe disease”. [2] The same effectiveness (and timeliness of PrEP availability), however, is not granted for future variants. Therefore, we reminded the “hypothetical” role of our recommendations: see below the changes:
- See Abstract Background: Pre-exposure passive immune prophylaxis (PrEP) may might contribute to improve HM outcomes
- See Abstract Results: Based on such a framework, the experts deemed that, was a variant-specific PrEP promptly available, it PrEP is would be considered mandatory for all candidates receiving allogeneic hematopoietic cell transplantation, CAR-T therapy, or bispecific antibodies, regardless of local viral epidemiology. PrEP was indicated During epidemiological waves variant-specific PrEP would be recommended also for patients with HMs at high risk of unfavorable COVID-19 clinical outcome
- See Discussion line 285. The available evidence suggests that in addition to the vaccination strategy, a variant-specific and promptly available PrEP is would be the optimal approach to contain the effects of COVID-19 in immunocompromised patients
Comment 4. They need to expand on how treatment-delay data (largely absent in literature) were inferred, this assumption is central but underdeveloped.
Response 4. The authors reckoned the difficulties in supporting “treatment delay” as the core determinants of PrEP decision. However, this is also the case for several clinical decisions. Moreover, treatment delay was not the unique criterion for determinants ranking. Therefore, we highlighted in a specific Discussion paragraph the need for clinical studies targeting specific outcomes of COVID19 prevention in oncologic patients. We also highlighted this limitation in the abstract.

Reviewer 2 Report
Comments and Suggestions for Authors
The manuscript proposes “evidence-based” recommendations for the use of passive pre-exposure prophylaxis (PrEP) against SARS-CoV-2 with anti–spike monoclonal antibodies in patients with hematologic malignancies, defining prescribing priorities through a combination of a systematic literature review and expert consensus using AHP and a modified Delphi process. Already in the abstract, PrEP is presented as potentially improving clinical outcomes, with the key deliverable being the designation of PrEP as “mandatory” for candidates for allogeneic HCT, CAR-T therapy, and bispecific antibodies “regardless of local epidemiology,” under the assumptions of “adequate vaccine uptake” and “PrEP effectiveness in reducing mortality/hospitalization.”
The topic is clinically relevant: patients with hematologic malignancies continue to exhibit an increased risk of severe COVID-19 and low seroconversion rates after vaccination, making PrEP a practical question. However, the current version does not read as methodologically mature. Core recommendations rely heavily on assumptions and expert-derived scales, while the computational framework underlying AHP is insufficiently transparent, the operational meaning of “evidence-based” remains diffuse, and there are internal inconsistencies between the declared dependence on epidemiologic context and the categorical stance of “mandatory regardless of epidemiology.”
Major concerns:
- Although the work is framed as evidence-based, the linkage between specific sources/levels of evidence and the final priority classes is frequently not traceable. Readers are presented with end-stage categories, but not with a transparent mechanism by which heterogeneous evidence streams (meta-analyses, observational studies, conference abstracts) were integrated into a single normative recommendation.
- There is an internal conflict of principles: “epidemiologic context matters” versus “mandatory irrespective of epidemiology.” On one hand, the authors emphasize that PrEP should be tailored to local epidemiology and viral variants; on the other, they assert absolute obligatoriness of PrEP for selected groups regardless of the epidemiologic scenario. This renders the algorithm difficult to interpret as a practical clinical rule and risks misapplication in real time, particularly when neutralizing activity of specific monoclonals is reduced.
- The endpoint of “treatment delay” is elevated to a key decision driver without direct empirical validation of the causal chain “PrEP prevents delays.” The text acknowledges the absence of published studies linking PrEP use to prevention of delays in antineoplastic therapy, yet treatment delays become a central rationale for strong recommendations. This is logically fragile and clinically risky, because “delay” is a complex composite influenced by organizational constraints, testing policies, and local protocols.
- The AHP/Delphi computational pipeline is insufficiently transparent and the reported metrics are not internally harmonized. The use of a 5-point relevance scale, subsequent derivation of “priority scores,” and introduction of “sr” with wide ranges appear to be methodologically heterogeneous elements that are not consolidated into a reproducible scheme. In the Supplementary material, “sr” effectively resembles a mean-to-SD ratio, which may overweight agreement (low variability) rather than clinical importance.
- Threshold-based classes and reported ranges suggest potential arithmetic and/or logical inconsistencies. Definitions of class boundaries and the stated ranges of risk scores do not consistently align, including instances where a class range includes values below the stated threshold. This is not a cosmetic issue, as these thresholds are precisely what translate expert ratings into clinical “mandatory/recommended” language.
- There is a risk of overclaiming PrEP effectiveness and generalizing recommendations to the contemporary landscape of variants and products. In several formulations, PrEP effectiveness in reducing severe outcomes is embedded as an assumption/constant, while the manuscript simultaneously emphasizes that clinical utility depends on the circulation of variants with reduced neutralization. Without a clear separation between “historical evidence” and “current applicability,” the recommendations risk becoming outdated at the time of publication.
- The systematic review component appears insufficiently rigorous for a manuscript making normative recommendations. Even with Supplementary material, concerns remain regarding bias (language restriction, potential publication bias), the lack of a clearly uniform and explicit approach to risk-of-bias appraisal across heterogeneous sources (especially abstracts), and how evidence of differing reliability was weighted in the final classification.
- Transparency regarding sponsorship and medical writing support is inadequate for a consensus-style recommendations paper. The combination of an “unconditional grant” and external medical writing support alongside a declaration of no conflicts of interest is, at minimum, ambiguous in the context of clinical recommendations on mAb prophylaxis and may undermine confidence in the neutrality of the consensus.
Author Response
The manuscript proposes “evidence-based” recommendations for the use of passive pre-exposure prophylaxis (PrEP) against SARS-CoV-2 with anti–spike monoclonal antibodies in patients with hematologic malignancies, defining prescribing priorities through a combination of a systematic literature review and expert consensus using AHP and a modified Delphi process. Already in the abstract, PrEP is presented as potentially improving clinical outcomes, with the key deliverable being the designation of PrEP as “mandatory” for candidates for allogeneic HCT, CAR-T therapy, and bispecific antibodies “regardless of local epidemiology,” under the assumptions of “adequate vaccine uptake” and “PrEP effectiveness in reducing mortality/hospitalization.”
The topic is clinically relevant: patients with hematologic malignancies continue to exhibit an increased risk of severe COVID-19 and low seroconversion rates after vaccination, making PrEP a practical question. However, the current version does not read as methodologically mature. Core recommendations rely heavily on assumptions and expert-derived scales, while the computational framework underlying AHP is insufficiently transparent, the operational meaning of “evidence-based” remains diffuse, and there are internal inconsistencies between the declared dependence on epidemiologic context and the categorical stance of “mandatory regardless of epidemiology.”
Major concerns:
Comment 1. Although the work is framed as evidence-based, the linkage between specific sources/levels of evidence and the final priority classes is frequently not traceable. Readers are presented with end-stage categories, but not with a transparent mechanism by which heterogeneous evidence streams (meta-analyses, observational studies, conference abstracts) were integrated into a single normative recommendation.
Response 1. The evidence-to-recommendation process could not be strictly adherent to GRADE standards, because of several underreported outcomes, e.g. treatment delay, and to inconsistencies among COVID-19 waves. Therefore, the authors agreed with the reviewer that “evidence-based” was not an adequate label for the project: we delete this label from the manuscript Title.
Comment 2. There is an internal conflict of principles: “epidemiologic context matters” versus “mandatory irrespective of epidemiology.” On one hand, the authors emphasize that PrEP should be tailored to local epidemiology and viral variants; on the other, they assert absolute obligatoriness of PrEP for selected groups regardless of the epidemiologic scenario. This renders the algorithm difficult to interpret as a practical clinical rule and risks misapplication in real time, particularly when neutralizing activity of specific monoclonals is reduced.
Response 2. The experts deemed that top ranked determinants prompted PrEP irrespectively of epidemiology, while, lower rank determinants required strong epidemiology support before supporting PrEP. We included Figure 2 to highlight this core concept.
Comment 3. The endpoint of “treatment delay” is elevated to a key decision driver without direct empirical validation of the causal chain “PrEP prevents delays.” The text acknowledges the absence of published studies linking PrEP use to prevention of delays in antineoplastic therapy, yet treatment delays become a central rationale for strong recommendations. This is logically fragile and clinically risky, because “delay” is a complex composite influenced by organizational constraints, testing policies, and local protocols.
Response 3. The authors reckoned the difficulties in supporting “treatment delay” as the core determinants of PrEP decision. However, this is also the case for several clinical decisions. Moreover, treatment delay was not the unique criterion for determinants ranking. Therefore, we highlighted in a specific Discussion paragraph the need for clinical studies targeting specific outcomes of COVID19 prevention in oncologic patients. Nevertheless, the benefit-to-risk ratio of PrEP was strongly harmonized wither cent IDSA recommendations.
Commento 4. The AHP/Delphi computational pipeline is insufficiently transparent and the reported metrics are not internally harmonized. The use of a 5-point relevance scale, subsequent derivation of “priority scores,” and introduction of “sr” with wide ranges appear to be methodologically heterogeneous elements that are not consolidated into a reproducible scheme. In the Supplementary material, “sr” effectively resembles a mean-to-SD ratio, which may overweight agreement (low variability) rather than clinical importance.
Response 4
-
- The AHP/Delphi computational pipeline was indeed very difficult to be implemented in order to satisfy the experts needs of comprehensiveness and robustness. Mean-to-Standard-Deviation ratio overweighted agreement, but it was necessary due to the limited size of the Panel and the several limitations of direct evidence.
- New added legend to Figure 1 discloses the bias of sr toward agreement.
Comment 5. A Threshold-based classes and reported ranges suggest potential arithmetic and/or logical inconsistencies. Definitions of class boundaries and the stated ranges of risk scores do not consistently align, including instances where a class range includes values below the stated threshold. This is not a cosmetic issue, as these thresholds are precisely what translate expert ratings into clinical “mandatory/recommended” language.
Response 5. We agree that class boundaries were not precise, however, the AHP process involved a small number of participants and clinical relevance as well as feasibility of the recommendations needed to be granted. Therefore, we highlighted such limitations in the discussion and suggested also for external validation of the results.
Comment 6. There is a risk of overclaiming PrEP effectiveness and generalizing recommendations to the contemporary landscape of variants and products. In several formulations, PrEP effectiveness in reducing severe outcomes is embedded as an assumption/constant, while the manuscript simultaneously emphasizes that clinical utility depends on the circulation of variants with reduced neutralization. Without a clear separation between “historical evidence” and “current applicability,” the recommendations risk becoming outdated at the time of publication.
Response 6.
-
- The reviewer suggested to abolish PrEP effectiveness from the baseline assumptions. However, recently published evidence further supports the effectiveness of variant-adapted PrEP[1]. Indeed IDSA recently recommended Pemivibart pre-exposure prophylaxis to most of immunocompromised individuals “because they have the highest risk of inadequate immune response and progression to severe disease”. [2] The same effectiveness (and timeliness of PrEP availability), however, is not granted for future variants. Therefore, we reminded the “hypothetical” role of our recommendations: see below the changes:
- See Abstract Background: Pre-exposure passive immune prophylaxis (PrEP) may might contribute to improve HM outcomes
- See Abstract Results: Based on such a framework, the experts deemed that, was a variant-specific PrEP promptly available, it PrEP is would be considered mandatory for all candidates receiving allogeneic hematopoietic cell transplantation, CAR-T therapy, or bispecific antibodies, regardless of local viral epidemiology. PrEP was indicated During epidemiological waves variant-specific PrEP would be recommended also for patients with HMs at high risk of unfavorable COVID-19 clinical outcome
- See Discussion line 285. The available evidence suggests that in addition to the vaccination strategy, a variant-specific and promptly available PrEP is would be the optimal approach to contain the effects of COVID-19 in immunocompromised patient.
- The reviewer suggested to abolish PrEP effectiveness from the baseline assumptions. However, recently published evidence further supports the effectiveness of variant-adapted PrEP[1]. Indeed IDSA recently recommended Pemivibart pre-exposure prophylaxis to most of immunocompromised individuals “because they have the highest risk of inadequate immune response and progression to severe disease”. [2] The same effectiveness (and timeliness of PrEP availability), however, is not granted for future variants. Therefore, we reminded the “hypothetical” role of our recommendations: see below the changes:
Comment 7. The systematic review component appears insufficiently rigorous for a manuscript making normative recommendations. Even with Supplementary material, concerns remain regarding bias (language restriction, potential publication bias), the lack of a clearly uniform and explicit approach to risk-of-bias appraisal across heterogeneous sources (especially abstracts), and how evidence of differing reliability was weighted in the final classification.
Response 7. The manuscript does not suggest “normative” recommendations. Rather, the project explores a methodology framework for highlighting comprehensive hematology-specific statements. Moreover, the manuscript is not a systematic review of the literature: several meta-analyses were published and summarized by the umbrella review reported in the Supplementary material. Therefore, the reported criticisms are not fully applicable. Risk-of-bias assessment was not our goal, due to the study major aim. The study design, e.g. meta-analyses first, and sample size were the core quality criteria. Evidence was not directly linked to the recommendations such as in a GRADE framework, therefore, deep quality assessment of the available evidence did not radically influence experts ranking of the determinants.
Comment 8. Transparency regarding sponsorship and medical writing support is inadequate for a consensus-style recommendations paper. The combination of an “unconditional grant” and external medical writing support alongside a declaration of no conflicts of interest is, at minimum, ambiguous in the context of clinical recommendations on mAb prophylaxis and may undermine confidence in the neutrality of the consensus.
Response 8. CoI Statement updated.

Round 2
Reviewer 1 Report
Comments and Suggestions for Authors
The authors have improved the paper and it is ready to be published.